# Comprehensive Analysis of the Prognostic Value and Molecular Function of CRNDE in Glioma at Bulk and Single-Cell Levels

**DOI:** 10.3390/cells11223669

**Published:** 2022-11-18

**Authors:** Lairong Song, Xiaojie Li, Xiaoying Xu, Xulei Huo, Yi Zheng, Xiaomin Wang, Da Li, Junting Zhang, Ke Wang, Liang Wang, Zhen Wu

**Affiliations:** 1Department of Neurosurgery, Beijing Tiantan Hospital, Capital Medical University, Beijing 100071, China; 2China National Clinical Research Center for Neurological Diseases, Beijing 100071, China; 3Key Laboratory of Carcinogenesis and Translational Research (Ministry of Education), Department of Lymphoma, Peking University Cancer Hospital & Institute, Beijing 100089, China

**Keywords:** CRNDE, glioma, prognostic model, single-cell RNA sequencing, epithelial–mesenchymal transition

## Abstract

Colorectal neoplasia differentially expressed (CRNDE) is an oncogenic long noncoding RNA (lncRNA) overexpressed in diverse malignancies. Here, we comprehensively analyze the prognostic value and molecular function of CRNDE in glioma. Bulk RNA-sequencing data from The Cancer Genome Atlas (TCGA) and Chinese Glioma Genome Atlas (CGGA), and single-cell RNA-sequencing data from the Tumor Immune Single-Cell Hub (TISCH) were analyzed. Kaplan–Meier survival analysis was applied to verify the prognostic value of CRNDE. Then, a nomogram based on multivariate Cox regression was established for individualized survival prediction. Subsequently, the expression characteristic and biological function of CRNDE were analyzed at the single-cell level. Lastly, the effects of CRNDE on the proliferation and invasion of glioma cell were explored in vitro. We discovered that CRNDE was a powerful marker for risk stratification of glioma patients. Regardless of the status of IDH and 1p/19q, CRNDE could effectively stratify patients’ prognosis. The nomogram that incorporated the CRNDE expression was proved to be a reliable tool for survival prediction. In addition, epithelial–mesenchymal transition may be the most important biological process regulated by CRNDE, which was identified at both the bulk and single-cell levels. Moreover, CRNDE knockdown significantly inhibited the proliferation and invasion of glioma cell. Overall, CRNDE is a vital oncogene and may be a valuable supplement to improve the clinical stratification of glioma.

## 1. Introduction

Glioma is the most prevalent primary malignant tumor in the central nervous system [1]. Traditionally, diffuse low-grade and intermediate-grade gliomas (WHO Grades II and III) are considered to be lower-grade gliomas (LGGs), and WHO Grade IV gliomas are considered to be glioblastomas (GBMs) [2,3]. GBM is the most malignant glioma subtype. Even with maximal surgical resection followed by postoperative radiochemotherapy, the median overall survival (OS) of patients is 14 months, and the 5-year survival rate is less than 5% [4]. Generally, the prognosis of LGG is better than that of GBM. However, the prognosis and therapeutic sensitivity of LGG vary widely due to high interindividual heterogeneity [5]. Some key molecular markers, such as IDH mutation, chromosome 1p/19q codeletion, and TERT promoter mutation, have been identified and incorporated into routine clinical assessment, significantly improving the prognosis stratification of LGG patients [6], even though the prognosis of some LGG patients could not be accurately assessed with existing molecular markers. To improve the clinical management of glioma, a better understanding of tumor biology at the molecular level is urgently needed.

LncRNAs participate in multiple biological processes in cancers [7,8]. CRNDE is a multifunctional lncRNA first found in colorectal cancer, and was identified to promote the proliferation and invasion of diverse malignancies [9]. The expression of CRNDE is upregulated in a variety of tumors and it is the most upregulated lncRNA in glioma [10]. The molecular functions of CRNDE in gliomas have been investigated in several studies. Zheng et al. reported that CRNDE could function as a ceRNA to regulate the miR-186-XIAP/PAK7 axis to promote the proliferation and invasion of glioma stem cells [11]. Zhao et al. found that CRNDE enhanced the resistance of temozolomide chemotherapy by activating the PI3K/Akt/mTOR pathway in glioma [12]. However, the role of CRNDE in the clinical stratification of glioma has not been described in detail.

With the development of high-throughput techniques, gene expression profiling facilitated our understanding of the molecular changes underlying tumor development and evolution. Bulk RNA-sequencing (RNA-seq) has been widely used to explore the expression and function of tumor-associated genes. However, traditional bulk RNA-seq methods sequence a mix of millions of cells, which fail to detect single-cell heterogeneity. Advances in single-cell RNA-sequencing (scRNA-seq) technologies have provided an unprecedented view of the cellular heterogeneity in tumors. Although CRNDE is overexpressed in glioma tissue, the expression characteristic of CRNDE and its biological function at the single-cell level are still unclear.

In the present study, we identify the role of CRNDE in the clinical stratification of glioma using bulk RNA-seq data from TCGA and CGGA [13]. In addition, we reveal the expression characteristic and biological function of CRNDE at the single-cell level using scRNA-seq data from the TISCH database [14]. Furthermore, we explore the effects of CRNDE knockdown on the proliferation and invasion of glioma cell. Our study provides new insights into the role of CRNDE in the development and evolution of glioma.

## 2. Materials and Methods

### 2.1. Patients and Datasets

The bulk RNA-seq data of 614 gliomas (456 LGGs and 158 GBMs) from TCGA and 847 gliomas (512 LGGs and 335 GBMs) from CGGA were included in this study. Due to the wide variance in the outcomes of LGG patients, we constructed a prognostic model to predict their prognosis. A total of 456 LGGs from TCGA were selected to enter the training set and develop the predictive model. Then, 512 LGGs from CGGA were used as the external validation set. Patients with survival of less than 30 days were excluded from the model construction.

### 2.2. Bulk RNA-Seq Data Processing

The fragments per kilobase per million mapped reads (FPKM) values of RNA-seq data were transformed into transcripts per million (TPM) to eliminate statistical biases [15,16]. To reduce the systematic error between TCGA and CGGA datasets, the expression level of CRNDE was normalized to [0,1] according to the previously mentioned method [5].

To assess the prognostic value of CRNDE, we selected the 456 LGGs from TCGA as the training set to generate the optimal cutoff value using the receiver operating characteristics (ROC) curve for predicting 5-year survival. The patients were then stratified into high- and low-expression groups according to the cutoff value. The same cutoff value and grouping method were applied to the 512 LGGs of CGGA to confirm its robustness. Kaplan–Meier survival analysis was performed to assess the stratification ability of the risk grouping in different molecular subtypes of LGG.

### 2.3. Development and Assessment of the Predictive Nomogram

To establish the predictive nomogram, multivariate Cox regression combined with stepwise backward elimination was applied to select the best factors included in the model [17]. Calibration curves were selected to assess the performance of the nomogram. Time-dependent ROC curves were used to evaluate the predictive accuracy.

### 2.4. Gene Set Enrichment Analysis (GSEA)

Differential expression analysis was conducted between the high- and low-expression groups in TCGA and CGGA, respectively. Then GSEA was performed on the basis of cancer hallmark gene sets (h.all.v7.5.1.symbols, http://www.gsea-msigdb.org/gsea/index.jsp, accessed on 24 June 2022) using R packages “fgsea” and “clusterProfiler”. Gene sets with normalized enrichment score (NES) > 1.5 and adjusted *p*-value < 0.05 were considered significant.

### 2.5. scRNA-seq Data Processing of Glioma Samples

The scRNA-seq data of glioma samples were downloaded from the TISCH database (http://tisch.comp-genomics.org/, dataset ID: Glioma_GSE131928, accessed on 28 July 2022). A total of 5311 cells from three patients (MGH115, MGH124, MGH125) were included in the analysis. R package “Seurat” was used to process the scRNA-seq data. R package “harmony” was applied to remove the batch effect among different samples. To reduce dimensionality, about 2000 highly variable genes were selected for principal component analysis and then summarized using the T-distributed Stochastic Neighbor Embedding (tSNE) algorithm. To identify marker genes for each cluster, we used the FindClusters function of the Seurat package with a resolution of 0.6. Cell clusters were annotated to known cell types according to marker genes from the CellMarker database.

### 2.6. Gene Set Variation Analysis (GSVA)

GSVA was used to calculate the pathway activity score of each cell on the basis of the 50 hallmark pathways (h.all.v7.5.1.symbols) using R package “GSVA”. R package “limma” was applied to analyze the differences in pathway enrichment between different cell clusters.

### 2.7. Cell–Cell Communication Analysis

Cell–cell communication analysis was conducted on the basis of the expression of ligands and receptors using R package “CellChat”. The ligand–receptor interactions between two cell clusters were evaluated on the basis of a permutation test. We extracted significant ligand–receptor pairs with *p*-value < 0.01.

### 2.8. Single-Cell Regulatory Network Inference and Clustering (SCENIC) Analysis

To analyze the transcriptional regulation of CRNDE in glioma cells, SCENIC analysis was performed using R packages “SCENIC” and “RcisTarget”. We selected the 50 glioma cells expressing the highest levels of CRNDE and the 50 glioma cells expressing the lowest levels of CRNDE for comparison by SCENIC. Then, we verified the correlation between the expression of transcription factors (TFs) and CRNDE using bulk RNA-seq data from TCGA.

### 2.9. Cell Culture and Transfection

Human glioma cell lines U118 (from Grade IV glioblastoma) and SW1783 (from Grade III astrocytoma) were cultured in Dulbecco’s Modified Eagle Medium (DMEM; Gibco, NY, USA) supplemented with 10% fetal bovine serum (FBS; BI, Kibbutz, Israel), 100 IU/mL penicillin, and 100 μg/mL streptomycin. All cells were incubated at 37 °C in a humidified incubator with 5% CO_2_. For transient CRNDE silencing, 3 × 10^5^ glioma cells were seeded into 6-well plates and cultured overnight. Next, cells were transfected with antisense oligonucleotide (ASO) at a final concentration of 50 nM using lipo3000 (Invitrogen, CA, USA). ASO-CRNDE-1, ASO-CRNDE-2, and negative control ASO (ASO-NC) were chemically synthesized by Ribio Company (Guangzhou, China). Transfection efficiency was verified with real-time quantitative PCR (RT-qPCR). The ASO sequences are listed in Appendix A.

### 2.10. RT-qPCR

Total RNA extraction was completed using MiniBEST Universal RNA Extraction Kit (TaKaRa, Dalian, China) following the manufacturer’s instructions. The PrimeScript™ RT Master Mix (TakaRa) was used for the reverse of cDNA. RT-qPCR was performed using TB Green^®^ Premix Ex Taq™ II (TakaRa) on an ABI 7500 real-time PCR system (Applied Biosystems, Foster, CA, USA). CRNDE quantitation was normalized to endogenous controls (GAPDH, β-actin, and α-tubulin). The relative expression of CRNDE was calculated using the 2^–ΔΔCT^ method. The primer sequences are also listed in Appendix A.

### 2.11. Cell Proliferation Assay

Glioma cells were seeded into 96-well plates at a density of 8 × 10^3^ cells per well, with 4 replicates. Cell viability was detected with Cell Counting Kit-8 (CCK-8, Dojindo, Kumamoto, Japan) at 0, 24, 48, 72, and 96 h after transfection. Absorbance was measured at 450 nm with the Spark Microplate Reader (Tecan, Switzerland).

### 2.12. Cell Invasion Assay

Cell invasion assay was conducted using Transwell invasion chambers (Corning, NY, USA). For Matrigel coating, Matrigel (BD Biosciences, Bedford, MA, USA) was diluted in serum-free DMEM at a ratio of 1:10. Glioma cells were resuspended in 200 μL serum-free DMEM at a density of 5 × 10^5^ cells/mL and then seeded in the upper chamber. DMEM with 15% FBS was added to the bottom chamber as an attractant. Nonmigratory cells on the upper membrane surface were carefully removed after incubation for 24 h. Cells that had invaded to the lower surface of the membrane were fixed with 4% paraformaldehyde and stained with 0.1% crystal violet.

### 2.13. Statistical Analysis

All statistical analyses were performed using R software version 4.1.3 (Free Software Foundation, Boston, MA, USA). All experiments were repeated three times to avoid contingency. *p* values < 0.05 were considered statistically significant.

## 3. Results

### 3.1. The Expression of CRNDE in Glioma

Differential expression analysis of CRNDE was performed between 200 normal brain specimens from the GTEx Portal and 668 glioma samples from TCGA. Compared to normal brain tissue, the expression of CRNDE was significantly upregulated in glioma and showed an elevated trend with the increase of tumor grade (Figure 1A). In addition, the expression of CRNDE in IDH-wildtype LGG was higher than that in IDH-mutant LGG (IDH-wildtype 16.33 vs. IDH-mutant 3.22, *p* < 0.001, Figure 1B). Similarly, the expression of CRNDE in 1p/19q noncodeleted LGG was higher than that in 1p/19q codeleted LGG (1p/19q noncodeleted 7.54 vs. 1p/19q codeleted 1.60, *p* < 0.001, Figure 1B). This suggested that the expression of CRNDE was closely related with the risk factors of glioma.

### 3.2. Kaplan–Meier Analysis Based on the Expression of CRNDE in TCGA and CGGA

IDH mutation and 1p/19q codeletion are considered to be the most important molecular markers of glioma due to their significant prognostic value. To verify the prognostic stratification ability of CRNDE, we performed Kaplan–Meier survival analysis on different molecular subtypes of glioma. In the training phase, we selected the 456 LGGs from TCGA as the training set to generate the optimum cutoff value of CRNDE using the ROC curve for predicting 5-year OS (Appendix A). As a result, patients with CRNDE expression higher than 0.2757 were divided into the high-expression group, while those with CRNDE expression lower than 0.2757 were divided into the low-expression group. As expected, regardless of the status of IDH mutation and 1p/19q codeletion, patients with high expression of CRNDE had worse prognoses than those with low expression of CRNDE in both the training and external validation sets (Figure 2 and Figure 3). This indicates that CRNDE is a valuable supplementary marker for glioma.

### 3.3. Establishment of a Predictive Nomogram for OS of LGG Patients

The characteristics of LGG patients in the training and validation sets are presented in Table 1. Univariate and multivariate Cox analyses were applied to identify the significant prognostic factors of LGG in TCGA and CGGA datasets. After adjusting for potential risk factors, including patient’s age, tumor grade, IDH mutation, and 1p/19q codeletion, multivariate Cox analysis showed that the expression of CRNDE remained an independent prognostic indicator for OS of LGG (Figure 4). Moreover, CRNDE was still an independent prognostic indicator for OS when all LGG and GBM patients were included (Appendix A).

After backward selection using Akaike’s information criterion as a stopping rule, a predictive nomogram that incorporated the patient’s age, tumor grade, IDH mutation, 1p/19q codeletion, and CRNDE expression was established to predict the 3- and 5-year OS of LGG patients in the training set (Figure 5A). Calibration curves reveal that the nomo-gram was well-calibrated in both the training and external validation sets (Figure 5B,C). In addition, the predictive accuracy was verified using time-dependent ROC curves. The nomogram gave a 3-year AUC of 0.92 and 0.81, and a 5-year AUC of 0.86 and 0.80 in TCGA and CGGA, respectively (Figure 5D,E), indicating that it was a good predictor for LGG patients’ survival.

### 3.4. Biological Processes Associated with the Expression of CRNDE in TCGA and CGGA

To identify the potential biological processes associated with the expression of CRNDE, GSEA was performed between the high- and low-expression groups in TCGA and CGGA, respectively. By ranking the NES, hallmark epithelial–mesenchymal transition (EMT) showed the highest score in both datasets (Figure 6), suggesting a close relationship between CRNDE and EMT in glioma. In addition, some pro-oncogenic signaling pathways were significantly enriched with high NES in both datasets, such as G2M checkpoint, E2F targets, JAK/STAT3 pathway, and the TNFA/NFKB pathway.

### 3.5. Cell Type Identification Based on scRNA-Seq from the TISCH

The expression feature and biological function of CRNDE at the single-cell level were also explored. A total of 5311 cells from three glioma patients (MGH115, MGH124, MGH125) were divided into four major clusters, namely, glioma cells, myeloid cells, oligodendrocytes, and T cells (Figure 7A). The tSNE plot clustered in the patients showed that the batch effect was fully corrected (Figure 7B). The expression features of marker genes of the four major clusters are presented in Appendix A.

For further cell type identification depending on the marker genes of minor clusters, glioma cells were divided into four minor subtypes (neural-progenitor-like (NPC-like) glioma cells, oligodendrocyte-progenitor-like (OPC-like) glioma cells, mesenchymal-like (MES-like) glioma cells, and astrocyte-like (AC-like) glioma cells), and myeloid cells were divided into three minor subtypes (microglial cells, monocytes, and neutrophils) (Figure 7C). The previous research of Cyril Neftel et al. [18] was also an important reference for the identification of glioma cell subtypes in this study. The expression of CRNDE at the single-cell level was projected onto tSNE, in which the feature plot showed that CRNDE was specifically overexpressed in glioma cells (Figure 7D). The expression profile of marker genes of the nine minor clusters is presented in Figure 7E.

### 3.6. Biological Processes Associated with the Expression of CRNDE at the Single-Cell Level

Among the four minor subtypes of glioma cells, the expression level of CRNDE in AC-like glioma cells was the highest, while that in NPC-like glioma cells was the lowest (Figure 8A–C). To identify the potential biological processes associated with the expression of CRNDE at the single-cell level, we performed GSVA to calculate the pathway activity score of each cell on the basis of 50 hallmark pathways. A direct comparison of AC-like versus NPC-like glioma cells revealed EMT as the top enriched hallmark pathway in AC-like glioma cells (Figure 8D,E). Remarkably, this is in high accordance with the GSEA in bulk RNA-seq. Moreover, the JAK/STAT3 and TNFA/NFKB pathways were also enriched with high GSVA scores in AC-like glioma cells at the single-cell level.

### 3.7. Cell–Cell Communication Analysis at the Single-Cell Level

To determine the potential interactions among different cell types, we applied CellChat analysis to identify the critical features of cell–cell communication and predict possible signaling pathways. Circle plots show that AC-like glioma cells communicated with other cells more frequently than NPC-like glioma cells did, whether as a source or target (Figure 8F,G). Moreover, many ligand and receptor molecules of the involved signaling pathways were related to EMT, such as CD44 [19], THBS1/CD47 [20], PTN [21], PTPRZ1 [22], WNT/FZD3 [23], NOTCH [24,25], and FGFR [26]. We next present EMT-related ligand–receptor interactions using bubble plots (Figure 8H,I). Compared to NPC-like glioma cells, AC-like glioma cells with high CRNDE expression exhibited higher EMT-related communication activity.

### 3.8. The Transcriptional Regulation of CRNDE in Glioma

To analyze the transcriptional regulation of CRNDE at the single-cell level, SCENIC analysis was applied. A set of TFs were predicted to regulate the expression of CRNDE at the transcriptional level, mainly including RFX4, CEBPD, DLX2, CEBPB, AR, and YBX1 (Figure 9A). Next, we evaluated the correlations between the expression of TFs and CRNDE using bulk RNA-seq data from TCGA. The result shows that the expression of the six TFs was significantly positively correlated with the expression of CRNDE in glioma (Figure 9B).

### 3.9. Biological Functions of CRNDE in Glioma Cells

To further assess the effect of CRNDE on glioma cells, we knocked down the expression of CRNDE with ASO in U118 and SW1783 cell lines. Knockdown efficiency was verified with RT-qPCR (Figure 10A,B and Appendix A). CCK-8 assay shows that the proliferation of glioma cells was significantly inhibited by CRNDE silencing (Figure 10C,D). In addition, Transwell assay reveals that CRNDE knockdown suppressed the invasion of U118 and SW1783 (Figure 10E–G).

## 4. Discussion

Due to the high interindividual heterogeneity, the prognosis of glioma patients, especially LGG patients, varies widely. Although molecular markers such as IDH mutation and chromosome 1p/19q codeletion have been incorporated into routine clinical evaluation to stratify the prognosis of LGG patients, there is still room for further improvement. In this study, we found that CRNDE is a valuable prognostic marker for glioma patients, and a potential supplement to IDH mutation and 1p/19q codeletion. We further built a nomogram that incorporated the patient’s age, tumor grade, IDH mutation, 1p/19q codeletion, and CRNDE expression to accurately predict the survival of LGG patients. The performance of the nomogram was verified in an external validation set to ensure its robustness and reliability.

CRNDE is an oncogenic lncRNA that plays important roles in cancers, including solid tumors and hematological malignancies [27]. CRNDE participates in multiple biological processes, such as cell proliferation, invasion, metastasis, autophagy, and apoptosis [28]. For example, Ding et al. reported that CRNDE could bind to EZH2 and epigenetically silence the expression of DUSP5 and CDKN1A, thereby promoting the proliferation of colorectal cancer cells [29]. Li et al. found that CRNDE promoted the invasion of osteosarcoma cells by activating Notch1 pathway and epithelial–mesenchymal transition [30]. Wang et al. demonstrated that CRNDE functioned as ceRNA to promote the metastasis of pancreatic cancer cells by regulating the miR-384/IRS1 axis [31].

The molecular functions of CRNDE in glioma have also been reported in several studies. However, the role of CRNDE in the prognostic stratification of glioma has not been described in detail. IDH mutation and 1p/19q codeletion are the most important prognostic markers to distinguish different molecular subtypes of glioma [6]. To verify the stratification ability of CRNDE in different molecular subtypes of glioma, we performed Kaplan–Meier analysis in TCGA and CGGA. As expected, CRNDE could effectively stratify the prognosis of patients regardless of the molecular subtype of LGG, which indicates that CRNDE may be a valuable supplement to improve prognostic evaluation. Moreover, CRNDE showed better prognostic stratification ability compared to other gene markers. For example, H2BC12 could predict poor survival outcomes of glioma patients, but failed to stratify the prognosis of IDH-wildtype patients [32]. The expression of eIF3 subunits was associated with poor OS of glioma patients, but still failed to stratify the prognosis of IDH-wildtype patients [33]. This highlights the excellent prognostic value of CRNDE. In addition, the effect of CRNDE on glioma cells was assessed. Cellular functional experiments showed that CRNDE knockdown significantly inhibited the proliferation and invasion of U118 and SW1783. This further confirmed the vital role of CRNDE in glioma.

To identify the biological process most related to CRNDE in glioma, we performed GSEA using bulk RNA-seq data from TCGA and CGGA. Interestingly, hallmark EMT showed the highest NES in both datasets, indicating a close relationship between CRNDE and EMT in glioma. Furthermore, we analyzed the cancer hallmarks related to CRNDE at the single-cell level. Among the four minor subtypes of glioma cell identified in the scRNA-seq dataset, AC-like glioma cells expressed the highest level of CRNDE, while NPC-like glioma cells expressed the lowest. Then, GSVA was applied to calculate the pathway activity score of each cell in the two cell subtypes. Remarkably, the comparison of AC-like versus NPC-like glioma cells revealed that hallmark EMT was also the top enriched pathway in glioma cells with high CRNDE expression. EMT is a process of epithelial cells acquiring mesenchymal features that is associated with tumorigenesis, invasion, metastasis, and resistance to chemotherapy [34]. Considering the bioinformatic results at the bulk and single-cell levels, the potential relationship between CRNDE and EMT should be given much attention. In addition, the JAK/STAT3 and TNFA/NFKB signaling pathways were associated with high CRNDE expression at both the bulk and single-cell levels. The activation of the JAK/STAT3 and TNFA/NFKB axes was widely reported to induced increased invasive and metastatic ability via enhancing EMT in various cancers [35,36,37,38,39]. This indicates that the JAK/STAT3 and TNFA/NFKB axes may participate in the regulation of CRNDE on EMT.

CellChat can quantitatively analyze the intercellular ligand–receptor interactions from scRNA-seq data, and help us in understanding the global communications among cells [40]. To determine the effect of CRNDE on glioma-cell-related intercellular communications, we selected AC-like and NPC-like glioma cells for comparison. Circle plots showed that glioma cells with high CRNDE expression communicated with other cells more frequently, suggesting a positive correlation between CRNDE expression and intercellular interactions. EMT can be induced by various signaling pathways, including TGF-β, BMP, Wnt-β-catenin, NOTCH, Shh, and receptor tyrosine kinases [24]. EMT-related signaling molecules were also included in CellChat analysis. For example, CD44 promotes EMT by upregulating ZEB1 in oral cancer cells [41]. CD47 activates EMT through modulating E-cadherin and N-cadherin in ovarian carcinoma [42]. PTN disrupts calcium-dependent cell adhesion and initiates the EMT of glioma cells [21]. FZD3 can activate the EMT of osteosarcoma cells by promoting β-catenin transfer into the nucleus [23]. According to bioinformatic analysis from CellChat, EMT-related signaling molecules were more frequently activated in AC-like glioma cells with high CRNDE expression, further confirming the strong association between CRNDE and EMT.

Considering the important role of CRNDE in glioma, its transcriptional regulation also deserves attention. SCENIC is a computational method to infer and reconstruct gene regulatory networks from scRNA-seq data [43]. In this study, we applied SCENIC analysis to infer the upstream TFs that may regulate the expression of CRNDE. As a result, we identified six TFs with the highest predictive scores: RFX4, CEBPD, DLX2, CEBPB, AR, and YBX1. Further correlation analysis using bulk RNA-seq data showed that the expression of these six TFs was significantly positively correlated with the expression of CRNDE, suggesting that these six TFs might contribute to the high expression of CRNDE in glioma. Notably, the six TFs exert pro-oncogenic effects in various cancers [44,45,46,47,48], which provides new sights for understanding the molecular mechanisms of glioma progression.

## 5. Conclusions

In summary, high CRNDE expression was associated with a malignant phenotype of glioma and indicated poor prognosis for glioma patients. CRNDE could effectively stratify the prognosis of LGG patients regardless of their molecular subtypes, which indicated that CRNDE may be a valuable supplement to IDH mutation and 1p/19q codeletion. Moreover, the nomogram that incorporated the patient’s age, tumor grade, IDH mutation, 1p/19q codeletion, and CRNDE expression was a reliable tool for the prognosis of LGG patients. In addition, EMT may be the most critical biological process regulated by CRNDE, which was inferred from RNA-seq analyses at bulk and single-cell levels. Lastly, an in vitro experiment confirmed that CRNDE contributed to the proliferation and invasion of glioma cells. This study presented a comprehensive understanding of the role of CRNDE in glioma at the bulk and single-cell levels, which may be helpful in the individualized management of glioma patients.

## Figures and Tables

**Figure 1 cells-11-03669-f001:**
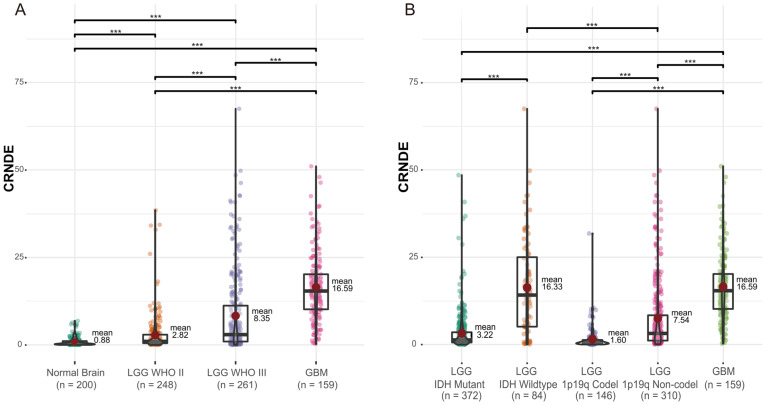
The expression of CRNDE in gliomas of different (**A**) grades and (**B**) molecular subtypes (*** *p* < 0.001).

**Figure 2 cells-11-03669-f002:**
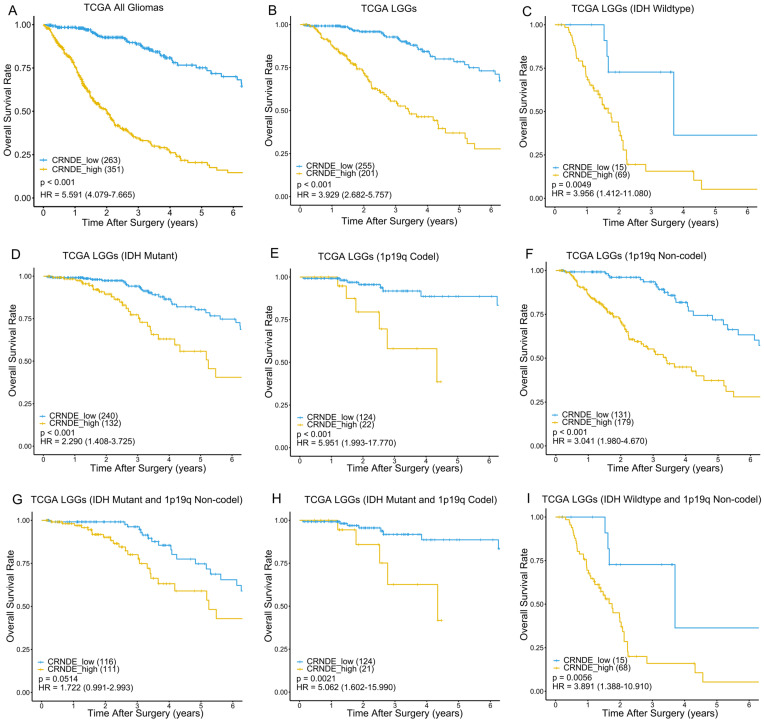
Kaplan–Meier survival curves of the high- and low-expression groups according to IDH and 1p19q status in TCGA dataset. Kaplan–Meier survival analyses were performed on (**A**) 614 glioma patients, (**B**) 456 LGG patients, (**C**) 84 patients with wildtype IDH, (**D**) 372 patients with IDH mutation, (**E**) 146 patients with 1p19q codeletion, (**F**) 310 patients without 1p19q codeletion, (**G**) 227 patients with IDH mutation and without 1p19q codeletion, (**H**) 145 patients with IDH mutation and 1p19q codeletion, and (**I**) 83 patients with wildtype IDH and without 1p19q codeletion.

**Figure 3 cells-11-03669-f003:**
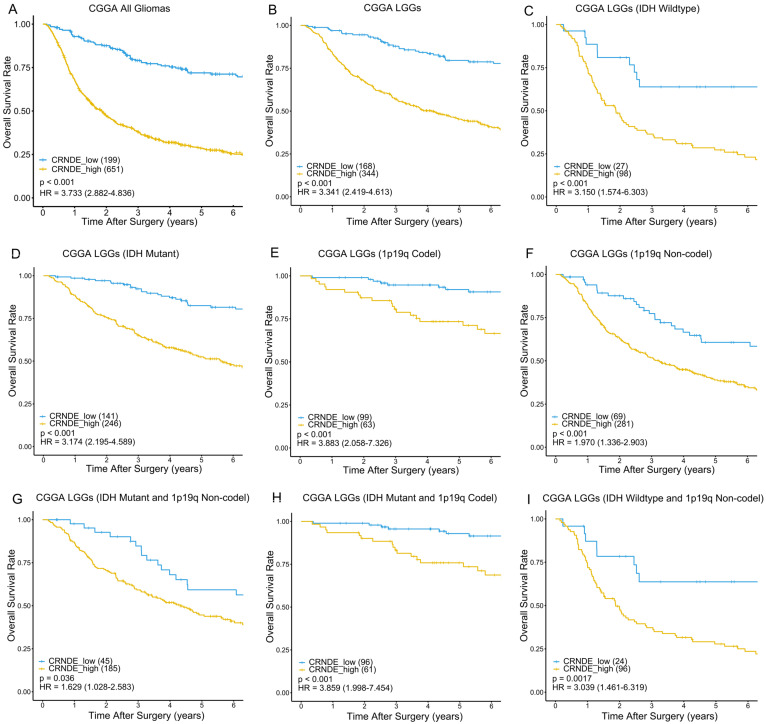
Kaplan–Meier survival curves of the high- and low-expression groups according to IDH and 1p19q status in CGGA dataset. Kaplan–Meier survival analyses were performed on (**A**) 850 glioma patients, (**B**) 512 LGG patients, (**C**) 125 patients with wildtype IDH, (**D**) 387 patients with IDH mutation, (**E**) 162 patients with 1p19q codeletion, (**F**) 350 patients without 1p19q codeletion, (**G**) 230 patients with IDH mutation and without 1p19q codeletion, (**H**) 157 patients with IDH mutation and 1p19q codeletion, (**I**) and 120 patients with wildtype IDH and without 1p19q codeletion.

**Figure 4 cells-11-03669-f004:**
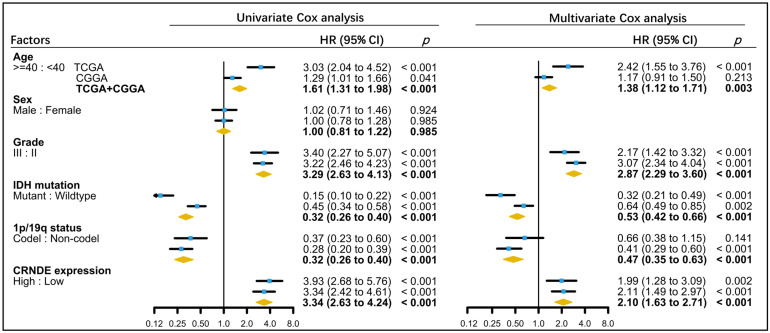
Association of clinical and molecular factors with OS based on univariate and multivariate Cox regression analyses of LGG patients in TCGA, CGGA, and the whole dataset.

**Figure 5 cells-11-03669-f005:**
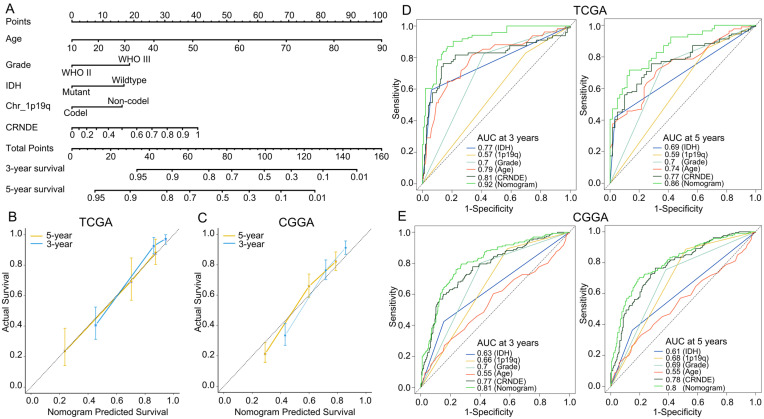
Nomogram model and its prediction performance. (**A**) Nomogram to predict the 3- and 5-year OS of LGG patients. (**B,C**) Calibration curves of the nomogram for predicting the 3- and 5-year OS in the (**B**) training and (**C**) external-validation sets. (**D,E**) Time-dependent ROC curves based on IDH mutation, 1p19q codeletion, tumor grade, patient’s age, CRNDE expression, and nomogram points in the (**D**) training and (**E**) external-validation sets.

**Figure 6 cells-11-03669-f006:**
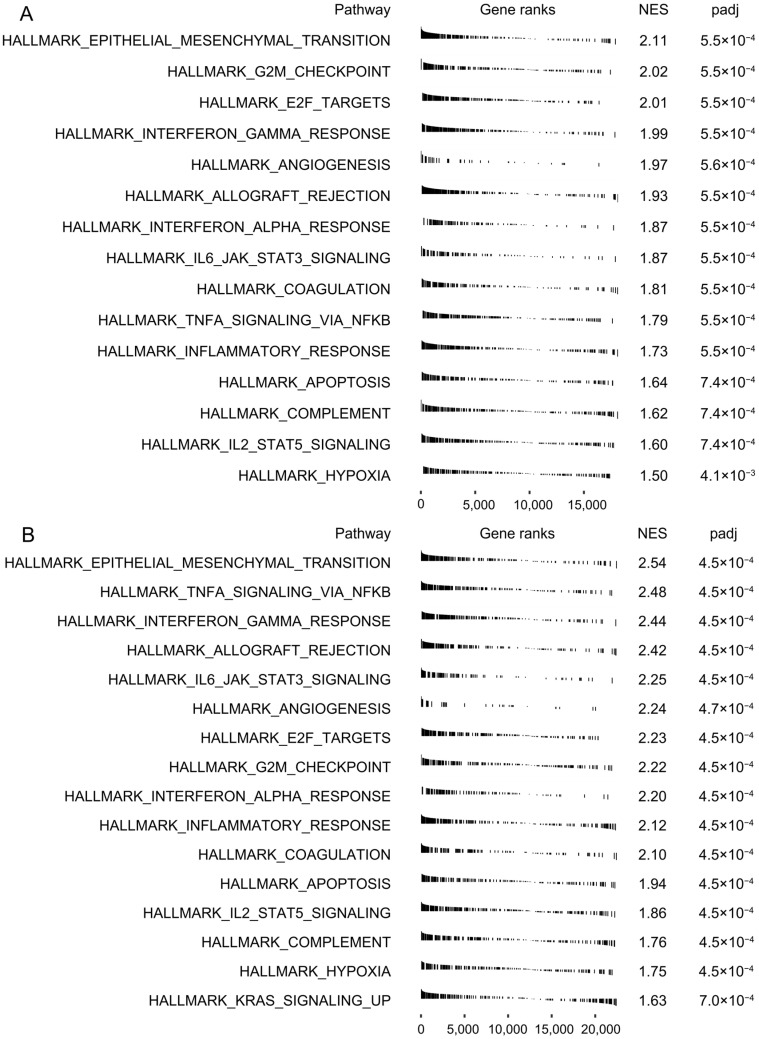
Biological processes associated with the expression of CRNDE in bulk RNA-seq. GSEA of hallmark pathways between the high- and low-expression groups in (**A**) TCGA and (**B**) CGGA.

**Figure 7 cells-11-03669-f007:**
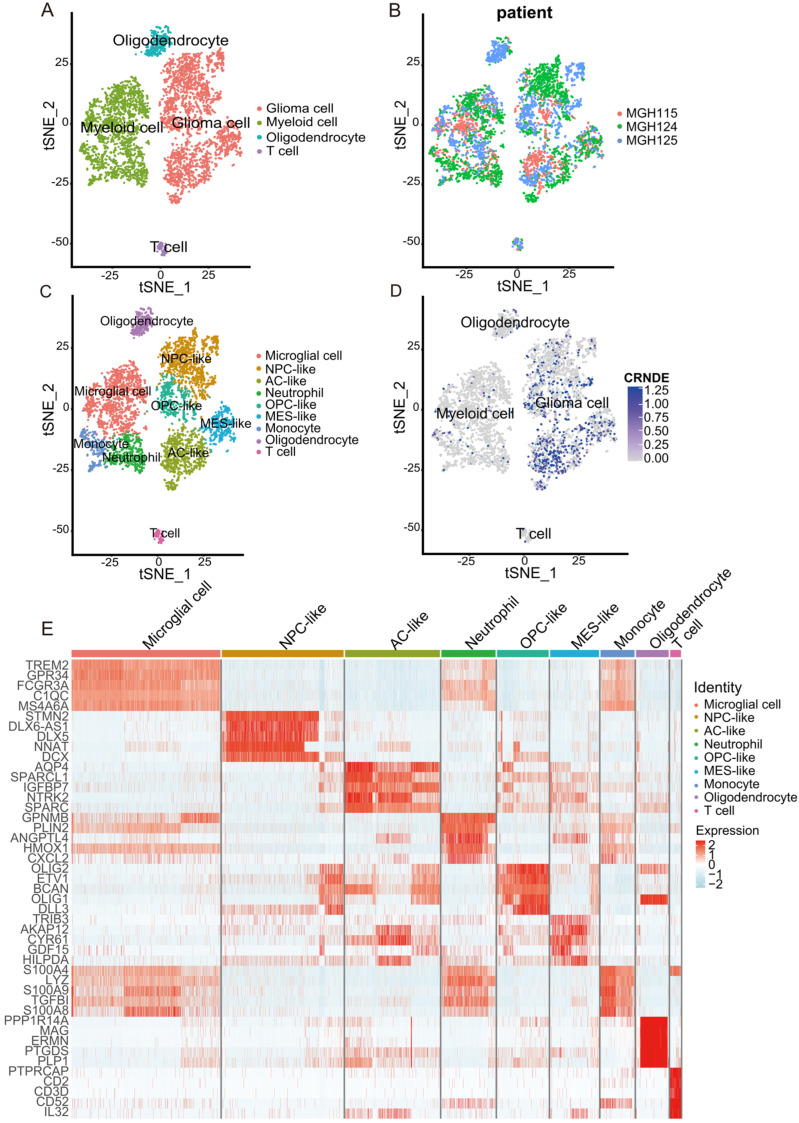
Overview of the scRNA-seq data from 3 glioma samples. (**A**–**D**) tSNE plot of the 5311 cells with each cell color-coded by (**A**) major cell type, (**B**) patient ID, (**C**) minor cell type, and (**D**) the expression level of CRNDE. (**E**) Heatmap of the top 5 marker genes for each cell type.

**Figure 8 cells-11-03669-f008:**
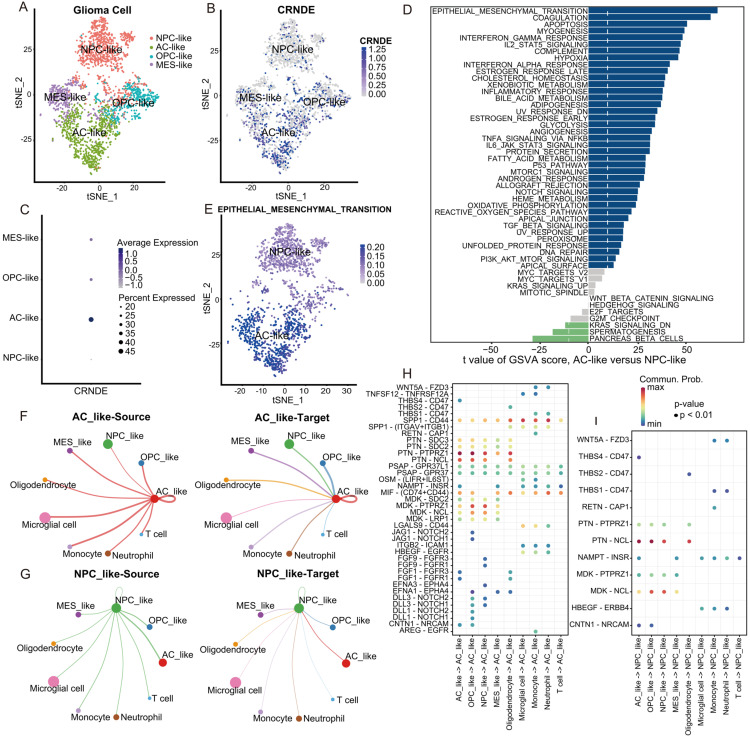
Biological processes and cell–cell communications associated with the expression of CRNDE at the single-cell level. (**A**,**B**) tSNE plot of the glioma cells with each cell color-coded by (**A**) minor subtype and (**B**) the expression level of CRNDE. (**C**) CRNDE expression in four minor subtypes of glioma cells quantified by bubble plot. (**D**) Differences in pathway activities scored per cell by GSVA between AC-like and NPC-like glioma cells. (**E**) tSNE plot of AC-like and NPC-like glioma cells with each cell color-coded by GSVA score of the EMT process. (**F**,**G**) The number of ligand–receptor interactions associated with (**F**) AC-like glioma cells and (**G**) NPC-like glioma cells. (**H**,**I**) EMT-related ligand–receptor interactions associated with (**H**) AC-like glioma cells and (**I**) NPC-like glioma cells.

**Figure 9 cells-11-03669-f009:**
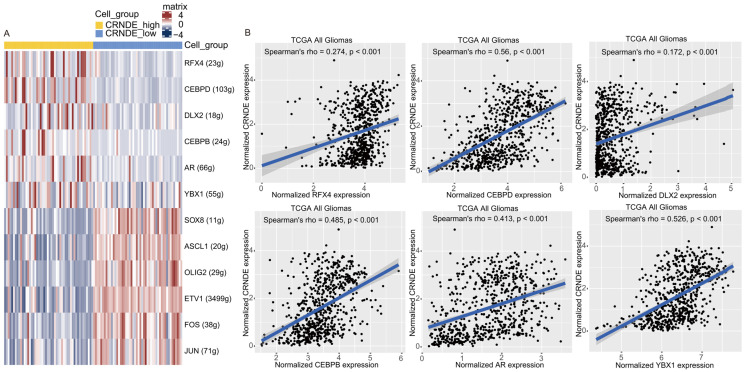
Transcriptional regulation of CRNDE in glioma. (**A**) Heatmap of TF activity by SCENIC in scRNA-seq data. (**B**) Correlations between the expression of CRNDE and RFX4, CEBPD, DLX2, CEBPB, AR, or YBX1 verified by Spearman’s correlation analysis in TCGA.

**Figure 10 cells-11-03669-f010:**
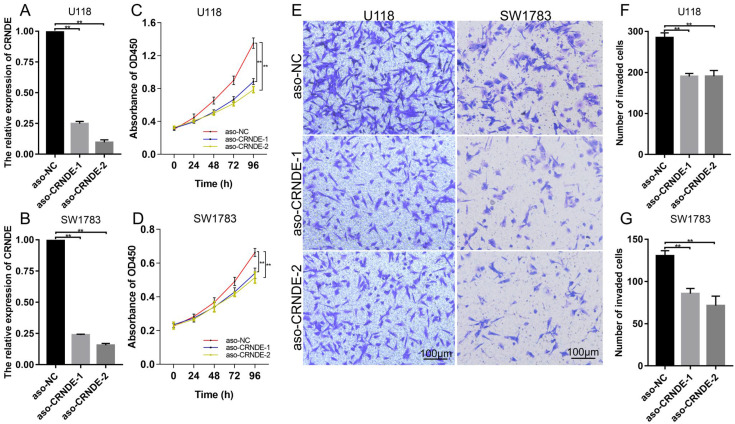
Effects of CRNDE knockdown on the proliferation and invasion of glioma cells. (**A**,**B**) Knockdown efficiency of CRNDE in U118 and SW1783 verified with RT-qPCR. The mean expression level of GAPDH, β-actin, and α-tubulin was used as the endogenous control. (**C**,**D**) Effect of CRNDE knockdown on the proliferation of glioma cells. (**E**–**G**) Effect of CRNDE knockdown on the invasion of glioma cells (** *p* < 0.01).

**Table 1 cells-11-03669-t001:** Characteristics of LGG patients in the training and validation sets.

TCGA LGGs		CRNDE Expression			CGGA LGGs		CRNDE Expression		
Factor	Number	High	Low	*p*	Factor	Number	High	Low	*p*
Total no. of patients	456	201	255		Total No. of patients	512	344	168	
Age (y)					Age (y)				
<40	210	71	139	<0.001	<40	242	151	91	0.031
≥40	246	130	116		≥40	269	192	77	
Sex					Sex				
Male	249	111	138	0.814	Male	296	205	91	0.243
Female	207	90	117		Female	216	139	77	
Grade					Grade				
WHO II	218	64	154	<0.001	WHO II	237	141	96	<0.001
WHO III	238	137	101		WHO III	275	203	72	
IDH					IDH				
Mutant	372	132	240	<0.001	Mutant	387	246	141	0.002
Wildtype	84	69	15		Wildtype	125	98	27	
1p19q					1p19q				
Codel	146	22	124	<0.001	Codel	162	63	99	<0.001
Noncodel	310	179	131		Noncodel	350	281	69	

## Data Availability

The data analyzed in this study can be found in online repositories. The names of the repositories and accession numbers are included in the article.

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
