# Peer review of "Comprehensive Analysis of the Prognostic Value and Molecular Function of CRNDE in Glioma at Bulk and Single-Cell Levels"

_cells, 2022, doi:10.3390/cells11223669_

Round 1

Reviewer 1 Report

This study has associated high CRNDE expression with a poor prognosis for glioma patients, and convincingly used expression to stratify the prognosis of LGG patients. Authors have eluded to the critical biological process regulated by CRNDE and performed some in vitro confirmation studies. The findings are clear.

- figure 1 is very crowded. Authors may consider representing the stats with * as they have with other graphs in the paper.

- CGGA and TCGA should be written in full the first time it is referenced.

- Since much of the work is reliant on an optimum cutoff value of CRNDE, authors should include, as supplementary figure, the ROC curve analysis used to obtain this value, and reference the figure on line 190.

Reviewer 2 Report

This is an article that has comprehensively profiled CRNDE in glioma using bulk as well as single-cell RNA-sequencing data. It is a nicely written article, easy to follow and comprehend. I only have few minor suggestions.

-       A p-value of exact zero is not possible. Please show us the exact p-value in scientific numbers.

-       It is quite difficult to read the figures in the current resolution and font size. Please improve the resolution and increase the font size. Also, please indicate which p-values were significant using asterisks. The level of significance should be mentioned in the figure legend.

-       Since many different datasets were used, it would be helpful if this is indicated either in the subheadings and/or figure legends, so that the readers can readily identify instead of having to read the body of the manuscript (i.e. bulk or single-cell etc).

-       Instead of only saying the expression level was “higher”, please also mention exactly with the level of significance. 

-       Full forms should be used before abbreviations are first introduced throughout the manuscript (i.e. gene names, lncRNAs, etc).

-       Correct terminology is RT-qPCR not qRT-PCR.

-       RT-qPCR result should be normalized to at least 3 endogenous controls. Please include 2 more and conduct the assay again. 

-       Figure 10A: Please show bars on top to indicate which values were compared to generate the p-values.

-       Figure 10E: Please show the entire images of the chamber. Also, please add a bar graph that quantifies the number of invaded cells, so that it is easier for readers to see the difference. Make sure to include p-values using all three experimental results.  

Round 2

Reviewer 2 Report

RT-qPCR bar graph should show the geometric mean of relative expression level using all 3 endogenous controls, not just GAPDH. The current graphs can be used as the supplementary figures. 
